# Preparation of Ceramsite Based on Waterworks Sludge and Its Application as Matrix in Constructed Wetlands

**DOI:** 10.3390/ijerph16152637

**Published:** 2019-07-24

**Authors:** Yaning Wang, Jinhu Yang, Hang Xu, Chenwei Liu, Zhen Shen, Kai Hu

**Affiliations:** 1College of Environment, Hohai University, Nanjing 210098, China; 2Ministry of Education, Key Laboratory of Integrated Regulation and Resource Development on Shallow Lakes, Hohai University, Nanjing 210098, China; 3Assistant Researcher, Rural Center, Nanjing Institute of Environmental Sciences of the Ministry of Ecology and Environment, Nanjing 210042, China

**Keywords:** waterworks sludge, constructed wetland, ceramsite

## Abstract

The recycling of waterworks sludge has become a trending issue because it not only solves the problem of difficult disposal but also saves land resources. This paper aimed to provide a new idea for the utilization of waterworks sludge to form ceramsite and to purify sewage. The specific surface area, average pore size, and pore volume of the made ceramsite were 8.15 m^2^/g, 8.53 nm, and 1.88 cm^2^/g, respectively. The made ceramsite was applied in a vertical-flow constructed wetland, and the removal efficiency of nitrogen, phosphorus and organic matter in sewage were investigated under the conditions of different start-up periods, hydraulic retention times, matrix filling heights and water quality. The removal rates of chemical oxygen demand (COD), ammonia nitrogen (NH_3_-N), and total phosphorus (TP) in the constructed wetlands were stable at 70%, 60%, and 79%, respectively. This constructed wetland with a ceramic matrix has certain advantages in the total amount of denitrifying microorganisms, with a proportion of 14.92%. The results prove the feasibility of preparing ceramsite from waterworks sludge and applying it as a matrix in a constructed wetland to purify sewage.

## 1. Introduction

Drinking waterworks sludge, which is the by-product of drinking water treatment processes, is considered water industrial waste. In China, the dry sludge amount reached 6.25 × 10^6^ tons in 2013, and the average annual growth rate was up to 13% from 2007 to 2013 [1]. Sludge usually consists of inorganic salts, heavy metals, pathogenic microorganisms and a small amount of organic contaminants [2]. If treated inappropriately, waterworks sludge can pose a serious threat to the ecological environment and human activities. Traditionally, waterworks sludge is mainly disposed of by sanitary landfill, incineration, ocean discharge, building materials and land applications [3]. However, no methods are ideal in terms of cost and application. In this sense, the harmlessness and resourcefulness of the use of the waterworks sludge has become a great challenge and has attracted widespread attention.

As a building material, ceramic granules are characterized by lightness, environmental protection, a low density, high strength, and high water absorption, with a bulk density of 1.2 g/cm^2^, a particle density of 2.6 g/cm^2^, a moisture content of 31%, and a compressive strength of 0.8 KN/seed [4]. The composition of waterworks sludge is almost the same as that of clay, so it can be used as a substitute for clay as a new source of raw materials in the preparation of ceramsite. The waterworks sludge can be dried and pulverized into powder and then mixed with other materials to be roasted into ceramsite [5]. Compared to slag and other fillers, ceramsite has better a removal of impurities and pollutants, with an increase of 8% for chemical oxygen demand (COD), 16% for ammonia nitrogen (NH_3_-N) and 25% for suspended solids (SS) from sewage [6]. Widely used in environmental protection, agriculture, horticulture, chemical industry and other fields, the application prospects of ceramsite are broad [7,8]. In addition, the use of ceramsite as a water treatment filler and microbial carrier will bring a meaningful reference to the actual production and development of sludge ceramsite [9,10].

The existing constructed wetland matrix is relatively single, and it has some shortcomings such as easy blockage, easy saturation, poor treatment effects, and a short operation cycle [11]. Therefore, developing and utilizing new and inexpensive materials to prepare composite substrates with a good adsorption effect, a long service life, and non-blocking should be the development direction of constructed wetland substrates [12].

The main objective of this paper was to study the feasibility of preparing ceramsite from the sewage sludge of a water supply plant and using it as a matrix in constructed wetlands. The removal rates of COD, NH_3_-N and total phosphorus (TP) were evaluated during start-up periods under the conditions of different hydraulic retention times, matrix filling heights, and water quality to examine the ability of the ceramsite applied in the constructed wetland to treat sewage. 

## 2. Methods

### 2.1. The Design Method of the Constructed Wetland

The constructed wetland experimental device is shown in Figure 1. It was basically a cylinder of organic glass material. Sewage was first distributed through the bottom water-distribution plate, and then it moved upwards through the waterworks sludge ceramsite substrate in the wetland system. Finally, the effluent was discharged from the top drainage pipe. The aim of the upward flow direction was to improve the internal environment of the matrix, make oxygen more sufficient to promote the proliferation of aerobic microorganisms, facilitate the better use of the wetland matrix to remove pollutants, and slow down matrix blockage [13]. The height of the ceramsite substrate was 600 mm, and the diameter was 50 mm. The total filling height was 750 mm in the device, and a certain distance was reserved for the overflow outlet. There were two wetland systems working in parallel.

To prevent the submerged leakage of the ceramsite and the distribution of the water, a 100 mm cobblestone supporting layer was placed on the bottom of the column on the 50 mm drilling plate. There were four rows of water intakes, and the heights were, respectively, 300, 450, 600, and 750 mm from the bottom. The experimental effluent was injected into the constructed wetland device through the peristaltic pump, and the hydraulic retention time (HRT) was adjusted by controlling the influent flow.

The constructed wetland system was in the start-up and adaptation period in the early stage of operation, and the effluent quality was unstable. Therefore, sampling and determination were carried out every other day. After the start-up and adaptation period, the effluent quality was stable, so sampling and determination were adjusted to twice a week. Referring to the standard of subsurface flow constructed wetland in “Technical Specification for Sewage Treatment of Constructed Wetlands” (HJ 2005–2010), the HRT of the constructed wetland during its start-up period was set to 2 days, and the hydraulic load was set to 0.122 m^3^/(m^2^·d). The start-up period was divided into two stages: The first stage operated for 30 days, using self-made water as the influent of constructed wetland. The COD was 300 mg/L, TN was 15 mg/L and TP was 3 mg/L. It was provided growth conditions which were suitable for the growth of microorganisms and promoted the growth and formation of biofilm so as to complete the start-up of constructed wetlands as soon as possible. The second stage operated for 20 days, and the influent used the tail water from Nanjing JX Sewage Treatment Plant to investigate the purification effect of the constructed wetland.

When the experimental period was from 0–50 d, which was the start-up period, the removal rate of COD, NH_3_-N and TP were tested. Then, from 51–110 d, different HRTs were selected, and the removal rate of COD, NH_3_-N and TP were again tested. Finally, from 111–130 d, the removal rate of NH_3_-N was tested when the influent NH_3_-N exceeded the standard. 

Different HRTs of sewage in the constructed wetland were controlled by adjusting the inflow rate, and then three HRTs for 1 d, 2 d, and 3 d were selected. After adjusting the HRT each time, water samples were collected after operating the constructed wetland device for 10 d. Then, with an HRT of 2 d, sampling and determination were adjusted to be taken every two days, and the removal rates of COD, NH_3_-N and TP were calculated at 15, 30, 45, and 60 cm, respectively. At each filling height, 10 samples were taken totally over the 20 d, and the mean value was calculated.

### 2.2. The Characteristics of Tail Water 

Table 1 is the tail water quality parameters of Nanjing JX Sewage Treatment Plant according to Class A of ‘Standard for Discharge of Pollutants from Urban Sewage Treatment Plant’ (GB 18918–2002) [14].

Because the concentration of pollutants in the effluent of the sewage treatment plant was at a medium level and the quality of effluent was representative, the effluent of the sewage treatment plant was taken as the following experimental influent.

### 2.3. The Detection Method of the Microbiological Indicator

The high-throughput sequencing work was commissioned by Shanghai Biotech Co., Ltd., which used the MiSeq (Illumina) platform for sequencing. On the 40th day of operation of the experimental device, a sufficient amount of waterworks sludge ceramsite was collected from the sampling port of the device, stored at minus 40 °C for 24 h, sealed in dry ice, and sent to the testing company. The testing company completed the sequencing work within 7 d.

### 2.4. Analytical Methods

The waterworks sludge ceramsite was characterized in terms of its morphology, structure, chemical composition and surface properties. A scanning electron microscope (SEM) (S4800, Hitachi Ltd., Tokyo, Japan) was used for analyzing the surface micromorphology and microstructure sizes of the prepared sludge ceramsite. The crystal structure and phase compositions were studied using X-ray diffraction (XRD) (THERMO 250XI, Thermo Fisher Scientific, Waltham, MA, USA). The Brunauer, Emmett and Teller (BET) specific surface areas and Barrett, Joyner and Halenda (BJH) pore size distribution were determined using a specific surface area-pore size analyzer (3H-2000PMI, Best Instrument Technology Co., Ltd., Beijing, China) by a nitrogen gas static adsorption method. The safety of the ceramsite was determined by its heavy metal leaching quantity. BET and pore size can directly reflect the adsorptive properties of materials. The nitrogen static adsorption method was used to determine the specific surface area and pore size. The samples were dried at 110 °C for 2 h, and then the nitrogen adsorption and desorption curves were measured at 77 K liquid nitrogen temperature. The specific surface area of the ceramsite from the feed water sludge was calculated by the BET multi-point method. The pore size distribution of ceramsite was analyzed by the BJH curve method. The existence of microporous structure of ceramsite was determined by the T-Plot method.

Other water quality indicators were detected by referring to “Water and Wastewater Monitoring and Analysis Method” (4th edition) [15]. COD was measured by potassium permanganate titration, NH3-N was measured by the Nessler reagent spectrophotometric method, total nitrogen (TN) was measured by potassium persulfate-ultraviolet spectrophotometry, and TP was measured by ammonium molybdate spectrophotometry.

## 3. Results and Discussion

### 3.1. Characteristics of Waterworks Sludge Ceramsit

The SEM images of the sludge ceramsite are shown in Figure 2, which reveals the surface morphology and structure size of the prepared ceramsite under different treatment conditions [16]. 

As shown in Figure 2, the raw ceramsite before roasting was mainly in the form of flakes or spots with fewer micropores. However, the roasted ceramsite formed more regular melts with developed internal voids. Therefore, the roasted ceramsite was more conducive to the removal of contaminants from the water. What is more, the large number of pore walls improved the attachment of the microorganisms and increased its bio-capacity.

Based on the data shown in Table 2, the specific surface area, average pore size, and pore volume improved significantly and increased by 34.27%, 37.14%, and 43.51%, respectively. 

High temperature roasting caused the hydroxyl and organic matter contained in the sludge to be removed. The organic matter volatilized out in the form of CO_2_ and the hydroxyl groups were transformed into water vapor at high temperatures, which played pore-forming roles. Therefore, the specific surface area, average pore size, and pore volume of the sludge ceramsite were all increased.

The specific surface area and pore size were determined using nitrogen adsorption–desorption, and Figure 3 shows nitrogen adsorption–desorption isotherms and pore size distribution of the ceramsite.

As shown in Figure 3a, the BET surface of the samples belonged to a type IV nitrogen adsorption–desorption isotherm, which meant that there was a mesoporous structure in the raw ceramsite and the roasted ceramsite [17]. The hysteresis loop of the isotherm tended to have a relative pressure equal to 1, indicating that there were large pores in the samples [18]. 

The pore size distribution of the raw sludge was irregular, as shown in Figure 3b. The pore size distribution of the raw ceramsite before roasting was irregular, most of which was concentrated at 5–40 nm, with a few greater than 40 nm. Conversely, the roasted ceramsite had distinct distribution characteristics which were based on micropores and mesopores, mostly concentrated at 0–20 nm. Thus, the roasted sludge ceramsite could be described as a mesoporous material. Additionally, the amount of micropores and mesopores in the roasted sludge ceramsite increased compared to that before roasting, which indicates that the roasted sludge ceramsite had a larger specific surface area and a better adsorptive desorption performance [19]. This was in agreement with the analysis results of the SEM.

### 3.2. Evaluation of the Application of the Constructed Wetland

In the process of wastewater treatment, COD removal should be the result of the combined action of matrix adsorption and microbial decomposition in constructed wetlands, but it should be mainly dependent on microbial degradation. Therefore, the effect of COD degradation largely depends on the number and growth of microorganisms in constructed wetlands. The removal effect of COD at the start-up stage showed that the ceramsite from feed sludge as the substrate of the constructed wetlands provided more attachment sites for microorganisms due to its rich microporous and mesoporous structure, which was beneficial to the growth and metabolism of microorganisms. 

The COD, NH_3_-N and TP removal efficiency during the start-up period in the first and second stages of the constructed wetland operation are shown in Figure 4. 

After 20 d of continuous operation in the first stage, the COD removal rate of the constructed wetlands reached a stable state of approximately 83%. This indicated that the biofilm of the constructed wetlands was basically mature and the film establishment period was over, and, as such, the test device could enter the normal operation stage. During the start-up period, the temperature was relatively suitable and the nutritional conditions for microbial growth were good, so the film-forming time was short. After the 31st day, the constructed wetland had entered the second stage of operation with an influent concentration of COD of approximately 35 mg/L. From Figure 4a, it can be seen that the removal rate of COD by the constructed wetlands decreased by approximately 70%, and the COD of the effluent was stabilized below 30 mg/L, which met the Class IV of Surface Water Environmental Quality Standard (GB3838-2002) [20].

From Figure 4b, it can be seen that the removal rate of NH_3_-N in the constructed wetlands first increased, then decreased slightly, and then finally tended to be stable. That is, the removal effect showed an overall trend of gradual improvement and then a slight decline. This is because in the early stages of the experiment, the constructed wetland was in an adaptive stage, the microorganisms on the substrate were in a growth stage, and the wetland system had not yet reached a stable state. At this time, the rapid adsorption of NH_3_-N by the ceramsite matrix of the feed sludge was the main method of NH_3_-N removal. As the experiment proceeded, the biofilm gradually formed and matured, and then the nitrification of the microorganisms became the dominant method of NH_3_-N removal. As the wetland ecosystem gradually became stable at the end, the removal rate of the NH_3_-N tended to be flat.

In the first stage of the operation, the removal rate of the NH_3_-N by the constructed wetland could reach 65%, while in the second stage, the influent concentration of NH_3_-N was maintained at approximately 2.1 mg/L. After reaching stable operation, the average removal rate of NH_3_-N was approximately 60%, and the effluent concentration was 0.8 mg/L, which met the standard of surface water type IV. Compared with other conventional wetland substrates, the removal efficiency of NH_3_-N was higher than that of steel slag and gravel (only 20%). Therefore, the ceramsite substrates of the feed sludge have certain advantages over other substrates.

During the start-up period, the removal efficiencies of TP in the first and second stages of the constructed wetland operation are shown in Figure 4c. The ceramsite contained a considerable amount of Al, Fe, and Ca elements, which had a high adsorption capacity for TP [21]. Therefore, the TP removal rate and effluent concentration of the constructed wetland were measured during the experiment to explore the practicability of using sludge ceramsite as a substrate for total phosphorus removal. Figure 4c shows that the removal rate of TP in the first stage of the start-up period was higher, while the wetland became gradually stable after 22 d with a removal rate of 85%. In the second stage, the influent concentration of TP was maintained at approximately 0.35 mg/L. Overall, the removal rate of TP was relatively stable (approximately 79%) but decreased a little compared with the first stage. The effluent concentration of TP met the Class IV of Surface Water Environmental Quality Standard (GB3838-2002) [20]. In the early stage of the operation, matrix adsorption was the main method of TP removal [22]. Comparing the removal effects of COD and NH_3_-N, it can be seen that the matrix adsorption had a greater impact on the removal effect of TP.

In addition, many studies have shown that among the various matrix materials, pure steel slag had the best removal rate of TP and could reach more than 90% [23], while gravel, zeolite, anthracite, and other matrix removal rates were only between 20% and 50% [24]. In contrast, the removal rate of total phosphorus by the ceramsite prepared in this study was much higher than that of most the matrix materials. The preparation of the ceramsite with sludge as the main raw material not only solves the problem of sludge disposal but also has a wide range of sources and a low price.

Hydraulic residence time (HRT) is an important parameter for constructed wetlands, and it can directly affect the removal efficiency of pollutants in water. Theoretically, the longer the hydraulic retention time, the better pollutant removal effect in constructed wetlands [25]. However, an excessive hydraulic retention time will cause a serious deflection of oxygen in a constructed wetland system. With an increasing hydraulic retention time, the treatment efficiency will decrease. What is worse, there exists the possibility that the pollutants are re-released into the water, which is not conducive to the purification of sewage [26,27]. On the other hand, it will also not achieve a good purification efficiency if it only pursues a low residence time.

This study focused on investigating the removal efficiency of COD, NH_3_-N and TP under different HRT conditions. During the experiment, the HRT of sewage in the constructed wetland was controlled by adjusting the inflow rate, and then three different HRTs for 1 d, 2 d, and 3 d were selected. The removal efficiency of the pollutants under different HRT conditions are shown in Figure 5.

As shown in Figure 5a, the COD concentration was between 35 and 40 mg/L, which experienced a slightly decrease, while the removal rate kept a rather stable increase of approximately 48%, 69%, and 73% with the extension of the HRT from 1 to 3 d, respectively. When the HRT reached 1 d, the effluent COD concentration was able to meet the IV water standard. The removal of COD in a constructed wetland system is mainly dependent on the metabolism of aerobic and anaerobic microorganisms. Therefore, a longer HRT provided sufficient time for the sewage to make contact between the ceramsite matrix and the biofilm of the feed sludge, as well as giving microorganisms chances to completely degrade organisms.

As shown in Figure 5b, the influent concentration of NH_3_-N fluctuated from 1.8 to 2.4 mg/L. Compared with the removal rate of COD, that of NH_3_-N fluctuated slightly with the extension of the HRT. When the HRT was 1 d, 2 d, and 3 d, the average removal rates of NH_3_-N in the constructed wetlands changed a little from 61% to 62% to 63%, respectively. The reason for this is that the removal of ammonia nitrogen from the constructed wetland system was mainly dependent on the role of the microorganisms. When the microbial membrane matured, the sewage could come in full contact with the substrate and its surface biofilm, and then NH_3_-N could be stably removed [28].

As shown in Figure 5c, the influent TP concentration fluctuated from 0.33 to 0.43 mg/L, while the removal rate of TP increased slowly at first and then decreased with the extension of the HRT. When the HRT was 1 d, 2 d and 3 d, the average removal rates of TP in the constructed wetlands were approximately 75%, 78%, and 70%, respectively. The effluent TP concentration was no more than 0.2 mg/L, which was much lower than the requirements of the surface water type IV standard. Under the conditions of the dynamic test, phosphorus in wastewater could be removed mainly by microbial assimilation. However, if the HRT was too long, it would create an anaerobic environment in the constructed wetland system, which is not conducive to biological phosphorus removal. In addition, too long of an HRT may lead to a matrix phosphorus desorption phenomenon, which leads to a decrease in the TP removal rate [29].

In summary, when the HRT reached 2 d or more, the ceramsite-based constructed wetland could effectively remove COD, NH_3_-N, and TP from the effluent. Considering that the hydraulic load should be as large as possible to reduce the occupied area of the constructed wetland, the HRT of the constructed wetland in this study should be 2 d. If the concentration of pollutants in the influent is high, the HRT can be prolonged to reduce the level of pollutants in the effluent.

The filling height of a constructed wetland matrix can directly affect its sewage purification effect and treatment cost. The higher the filling height, the better the sewage purification effect, and the cost also increases. Therefore, it is very important to choose an appropriate filling height. The optimum filling height of the ceramsite was determined according to a small-scale test. In this study, the removal rates of COD, NH_3_-N and TP in wastewater were investigated in the constructed wetland when the filling height of the ceramsite in the feedwater sludge was 15, 30, 45, or 60 cm. The treated wastewater was the same as the wastewater mentioned before in Table 1 with an HRT time of 2 d. The removal efficiency of pollutants at different filling heights is shown in Figure 6.

Though the data fluctuated above and below the mean value, the deviation was small and the overall trend can still clearly show the difference between COD, NH_3_-N and TP. With the increase of filling heights, the removal rate of COD rose continuously. Compared to the filling heights from 30 to 60 cm, that from 0 to 30 cm had more obvious growth for the removal rate. This showed that the removal of COD from the constructed wetlands concentrated in the bottom half of the filling height because the bottom half included the inlet of the device, and the carbon sources in the sewage were abundant so that the microorganisms multiplied rapidly and the biomass was abundant. Thus, the ability to remove pollutants was stronger. As the distance the sewage flows through the matrix became longer and longer, the nutrients in the water decreased continuously, which inhibited the growth of microorganisms and caused a decline in their removal ability [30]. In addition, with the increase of filling height, the load of organic matter in the sewage decreased gradually, so the removal rate of the COD was relatively low.

The removal rate of the NH_3_-N rose with the increase in the filling height, and, especially, when the filling height was 30–45 cm, the removal rate of NH_3_-N experienced its fastest increase. Different from that of COD, the removal rate of NH_3_-N in the constructed wetland mainly occurred in the top half of the filling height. When the filling height was 30 cm, the removal rate of NH_3_-N was only approximately 25%. The reason for this phenomenon was that oxygen was abundant in the bottom half of the filling height and aerobic heterotrophic bacteria metabolized vigorously and proliferated rapidly, which inhibited the growth of nitrifying bacteria. With the increase of the filling height, the aerobic heterotrophic bacteria gradually decreased, and nitrifying bacteria could grow and proliferate. Thus, the removal rate of NH_3_-N increased faster [31].

For TP, its removal rate also rose with the extension of the filling height. When the filling height was 0–30 cm, the increase rate was relatively faster, but it became slower in the latter half. This indicated that the first half of the constructed wetland had a better removal effect of TP, because the first half had richer nutrients and the reproduction of microorganisms were vigorous. In the actual operation process of a sewage plant, there will be unstable operations such as the illegal drainage of enterprises and process adjustments, so the quality of the effluent will not be stable all the time, which leads to an excessive concentration of influent water in constructed wetlands. Therefore, constructed wetlands must have a certain impact load resistance. During the actual operation of the sewage treatment plant, based on the existing research on constructed wetlands and the abovementioned experimental analysis, NH_3_-N is a water quality index that can easily exceed the standard. Therefore, this section investigated the removal efficiency of NH_3_-N from sewage by a ceramsite-based constructed wetland with feedwater sludge when the influent NH_3_-N of the constructed wetland exceeded the standard. Figure 7 shows that the ceramsite-based constructed wetland with feedwater sludge could resist the impact of increasing NH_3_-N concentration in the influent to a certain extent.

When the NH_3_-N concentration in the influent exceeded the design value (2.19 mg/L), for example 5.3–5.7 mg/L, the NH_3_-N concentration in the effluent was maintained at 2.3–2.6 mg/L, which still met the discharge standard of Class An of GB 18918–2002 [14].

### 3.3. Analysis of the Removal Mechanism

Microbial community diversity has the function of maintaining ecosystem stability. During the operation of the constructed wetlands, the higher the diversity of microbial communities, the better the stability of the constructed wetlands ecosystem. Table 3 shows the Alpha diversity index of the constructed wetland microorganisms in this study by doing high-throughput sequencing work.

As shown in Table 3, Chao1 (richness estimate for operational taxonomic units) and the ACE (another richness estimate for operational taxonomic units) analysis index can be used to calculate and analyze the distribution abundance of the microbial communities. The Shannon, Simpson and the Cverage indexes can analyze the diversity of the microbial community distribution [32]. A coverage value of 0.95 reflected the high coverage of samples and the low probability of undetected sequences. Therefore, the sequencing results of this experiment are representative and could represent the real situation in the wetlands. The larger the Shannon value, the higher the community diversity [33] but The larger the Simpson value, the lower its abundance [34]. From the data shown in Table 3, it can be seen that the directivity of the two indices were the same, which indicates that the diversity of the microbial community in the constructed wetland system was high in this study.

To further analyze the diversity of the microorganisms in the constructed wetlands, the relative abundance of the main species in the community was investigated. There were many species of microorganisms in the constructed wetlands at a generic level in this study. However, in this section, the first 50 species of microorganisms at the generic level were selected for analysis. The effective sequences at the microbial genera level in the constructed wetlands are shown in Figure 8.

From Figure 8, it can be seen that *Enterobacter*, *Geobacter* and *Loacibacterium* (Acidobacter) were relatively abundant. In the detection of denitrifying microorganisms, *Thiobacillus*, *Dechloromonas*, *Rhizobium*, *Azospirillum* and *Azospira* were found. *Rhizobium*, *Azospirillum* and *Azospira* are heterotrophic denitrifying bacteria, *Thiobacillus* is an autotrophic denitrifying bacteria, and *Dechloromonas* has a strong nitrate reduction ability [35]. Overall, the proportion of denitrifying bacteria in the constructed wetlands was 14.92%. Apart from that, *Zoogloea*, which are related to the formation of microbial micelles, were detected. *Zoogloea* is vital to promote simultaneous nitrification and denitrification processes, and, therefore, it plays a key role in sewage treatment. Finally, it can be seen that the ceramsite-constructed wetland had certain advantages in the total amount of denitrification microorganisms.

## 4. Conclusions

The study proves the feasibility of preparing ceramsite from the sewage sludge of a water supply plant and using it as a matrix in the constructed wetland to treat sewage. The specific surface area, average pore size, and pore volume of the prepared ceramsite are 8.15 m^2^/g, 8.53 nm, and 1.88 cm^2^/g, respectively. It can not only help solve the problem of sludge resource disposal from water supply plants but also effectively remove nitrogen, phosphorus, and organic matter with the removal rates of COD, NH_3_-N, and TP at 70%, 60%, and 79%, respectively. This constructed wetland with the ceramic matrix also has certain advantages in the total amount of denitrifying microorganisms with a proportion of 14.92%. It is suggested that sewage sludge can be combined with other raw materials to produce ceramsite, such as fly ash and river sediment, which not only saves the bentonite resources but also helps to turn waste into treasure. Additionally, the modified sludge ceramsite based on the sludge ceramsite can be developed for the removal of specific pollutants.

## Figures and Tables

**Figure 1 ijerph-16-02637-f001:**
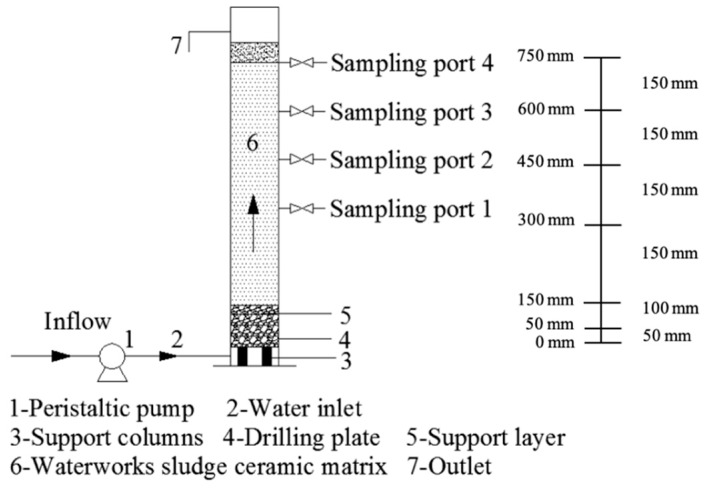
Schematic diagram of the experimental device for the constructed wetland.

**Figure 2 ijerph-16-02637-f002:**
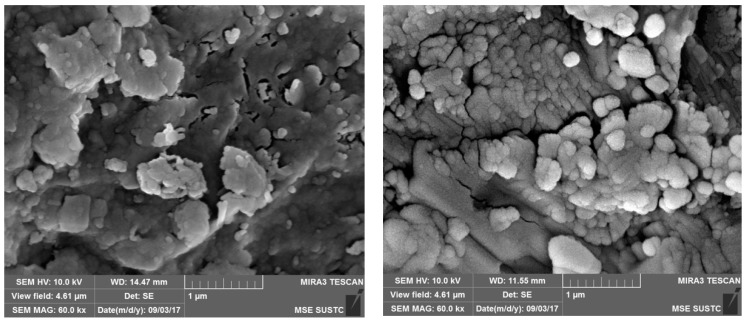
SEM images of the sludge ceramsite before (**left**) and after (**right**) it was roasted.

**Figure 3 ijerph-16-02637-f003:**
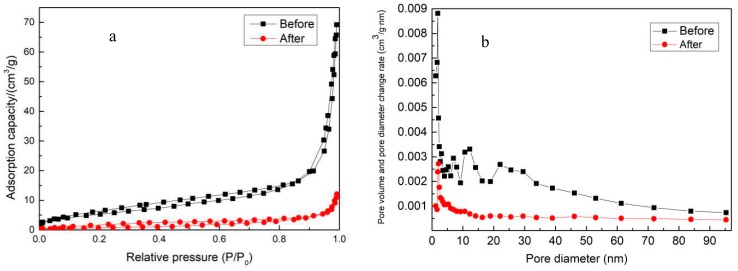
Nitrogen adsorption–desorption isotherms (**a**) and pore size distribution (**b**) of ceramsite.

**Figure 4 ijerph-16-02637-f004:**
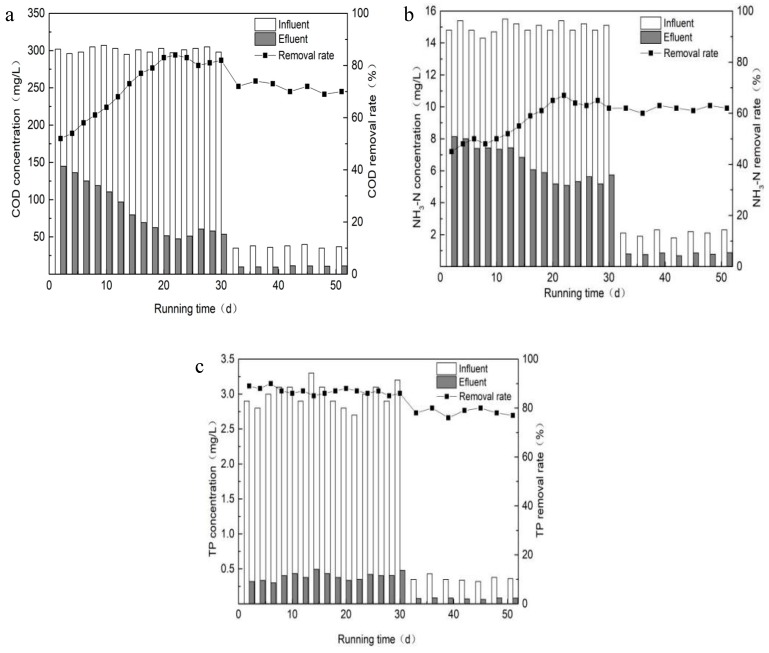
Chart of chemical oxygen demand (COD) (**a**), ammonia nitrogen (NH_3_-N) (**b**), and total phosphorus (TP) (**c**) removal during the start-up stage.

**Figure 5 ijerph-16-02637-f005:**
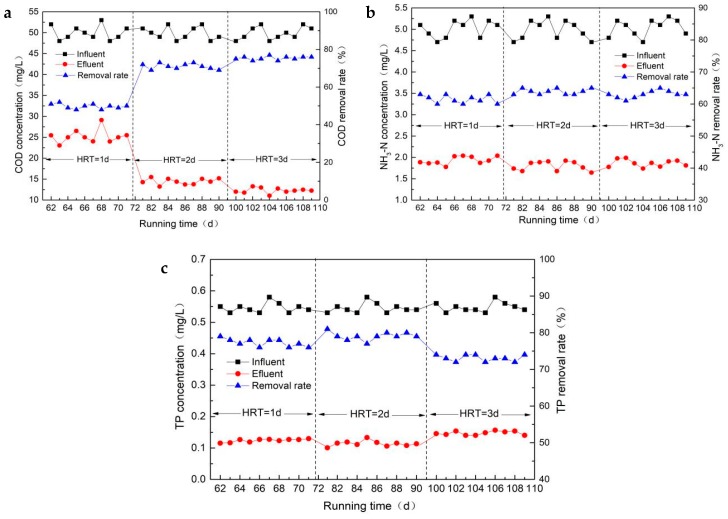
Chart of COD (**a**), NH_3_-N (**b**), and TP (**c**) removal under different hydraulic retention times (HRTs).

**Figure 6 ijerph-16-02637-f006:**
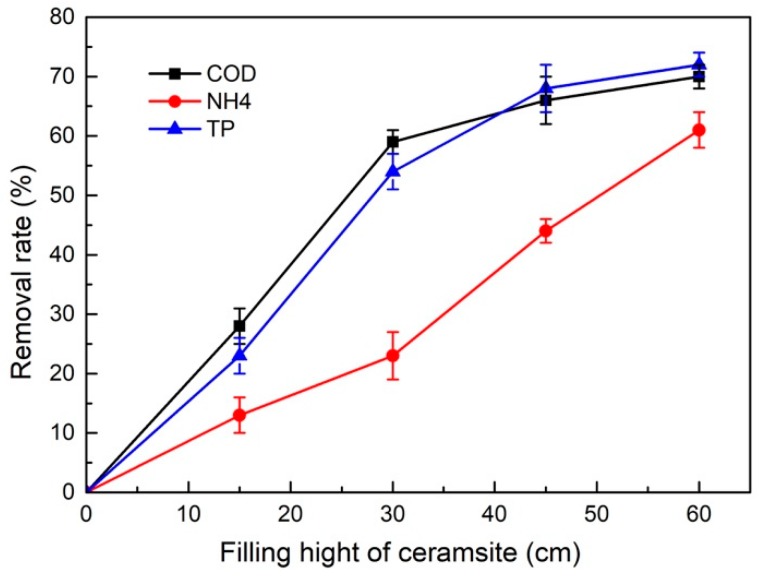
Chart of COD, NH_3_-N and TP removal at different filling heights.

**Figure 7 ijerph-16-02637-f007:**
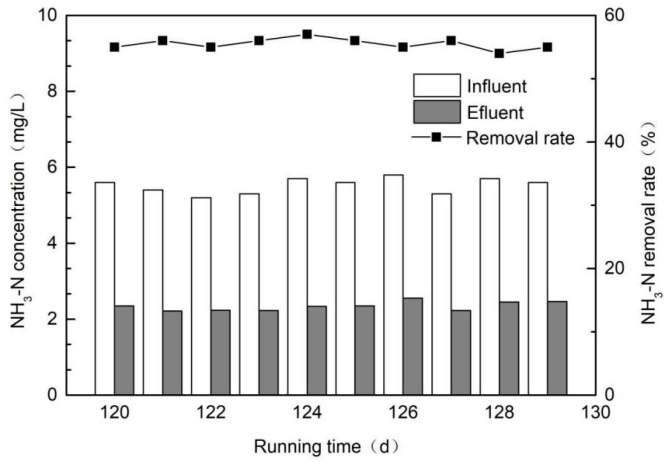
Chart of NH_3_-N removal when the influent NH_3_-N exceeded the standard.

**Figure 8 ijerph-16-02637-f008:**
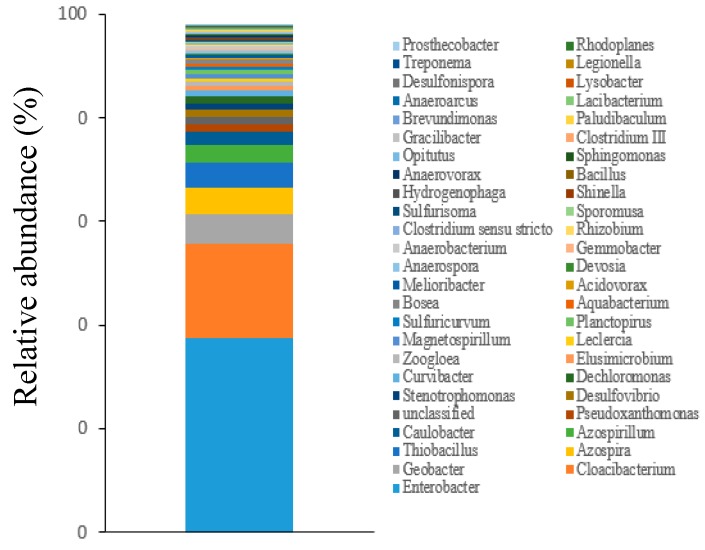
Population abundance map at the genus level of microorganisms.

**Table 1 ijerph-16-02637-t001:** Water quality index of tail water from Nanjing JX Sewage Treatment Plant.

pH	COD (mg/L)	NH_3_-N (mg/L)	NO_3_-N (mg/L)	NO_2_-N (mg/L)	TN (mg/L)	TP (mg/L)
7.4	38.24	2.19	9.35	0.67	13.06	0.35

**Table 2 ijerph-16-02637-t002:** The physical properties of the waterworks sludge ceramsite.

Samples	Specific Surface Areas (m^2^/g)	Average Pore Size (nm)	Pore Volume (cm^3^/g)
Raw ceramsite	6.07	6.22	1.31
Roasted ceramsite	8.15	8.53	1.88

**Table 3 ijerph-16-02637-t003:** Alpha diversity index of microorganisms.

Index	OTU	Shannon	Simpson	Coverage	ACE	Chao1
Number	3034	3.04	0.18	0.95	125,197.23	41,587.38

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
