# Peer review of "Preparation of Ceramsite Based on Waterworks Sludge and Its Application as Matrix in Constructed Wetlands"

_ijerph, 2019, doi:10.3390/ijerph16152637_

Round 1
Reviewer 1 Report
The contribution use a vertical-flow constructed wetland with ceramsite made from waterworks sludge. In the methods section, the authors did not include the type of wastewater to be treated, nor its characteristics. It is required that the authors cite previous works related to the use of ceramsite or similar materials used in wetlands Bibliographic references are required Page 1, Line 42: Widely used in environmental protection, agriculture, horticulture, chemical industry and other fields, the application prospects of ceramsite are broad. Page 3, line 82: "A certain amount" How much? Page 4, line 117: "What's more" Rephrase In the text Figure 3a and 3b are mentioned but in the figure there is no "a" and "b". It is also mentioned in Figure 3 "before" and "after" without explanation in the text. Page 5, line 138 Rephrase "obvious" Page 5, line 156 "Fogure" Page 7, line 208 Error! Reference source not found.Author Response
Response to Reviewer 1 Comments
Point 1: In the methods section, the authors did not include the type of wastewater to be treated, nor its characteristics.
Response 1: Include something about characteristics of wastewater
2.2. The characteristics of wastewater
Table 1. Water quality index of wastewater from Nanjing JX Sewage Treatment Plant.
pH | COD(mg/L) | NH3-N(mg/L) | NO3-N(mg/L) | NO2-N (mg/L) | TN (mg/L) | TP (mg/L) |
7.4 | 38.24 | 2.19 | 9.35 | 0.67 | 13.06 | 0.35 |
Table 1 is the tail water quality parameters of Nanjing JX Sewage Treatment Plantaccording to Class A of ‘Standard for Discharge of Pollutants from Urban Sewage Treatment Plant’(GB 18918-2002). Because the concentration of pollutants in the effluent of the sewage treatment plant is at a medium level and the quality of effluent is representative, the effluent of the sewage treatment plant is taken as the following experimental influent.
Point 2: It is required that the authors cite previous works related to the use of ceramsite or similar materials used in wetlands Bibliographic references are required Page 1, Line 42: Widely used in environmental protection, agriculture, horticulture, chemical industry and other fields, the application prospects of ceramsite are broad.
Response 2: Add an reference
[6] Wang, X.X.; Xu, L.L.; Yang, Y.L. Experiment study on carrier performance of bio-contact oxidation. Shanxi Architectural 2004, 23, 98-99.
Point 3: Page 3, line 82: "A certain amount" How much?
Response 3: Delete ‘The gravimetric method was used to examine the growth of the biomass on the surface of the ceramsite. A certain amount of ceramsite was placed in a crucible, rinsed with distilled water and dried in an oven at a temperature of 105°C until its weight was constant. The weight W1 was recorded. Then, we added a 5% NaOH solution to the dried ceramsite and continued to heat it with stirring until the biofilm was completely detached. Then, we repeated the above method and recorded the weight W2. The amount of biomass attached to the ceramsite was (W1−W2)/W2 (g VSS/g ceramsite).’
Point 4: Page 4, line 117: "What's more" Rephrase In the text Figure 3a and 3b are mentioned but in the figure there is no "a" and "b".
Response 4: Add the "a " and "b" to the Figure 3.
Point 5: It is also mentioned in Figure 3 "before" and "after" without explanation in the text. Page 5, line 138.
Response 5: "Additionally, the amount of micropores and mesopores in the roasted sludge ceramsite increased compared to that before roasting, which indicates that the roasted sludge ceramsite has a larger specific surface area and a better adsorptive desorption performance. " is the explanation for "before" and "after" .
Point 6: Rephrase "obvious" Page 5, line 156 "Fogure" Page 7, line 208 Error! Reference source not found.
Response 6: Change obvious to distinct, Fogure to Figure, the reference in line 208 do exist.
Vymazal J. Constructed wetlands for treatment of industrial wastewaters: A review. Ecological Engineering 2014, 73, 724-751.
Reviewer 2 Report
Dear Authors. The subject of the research is interesting. However, in my opinion, your work is not about a treatment wetland, because the treatment wetland concept includes the plants use. In your work, you are talking about a way to improve the medium for the plants in a vertical up-flow treatment wetland. Please consider this in your title. Furthermore, the text was written in a poorly way. Main problems are related to the description of materials and methods section. In the present version, you did not describe in a detailed way your materials an methods. For this reason, the reading of the results and discussion is hard to follow. In addition, you have serious problems with the final edition of the text. To next, you can find several comments about your paper:
Line 17: You want to add “…were evaluated”, is correct?
Line 37: Can you give some numbers related with the properties described by you?
Line 42: What is the enhanced given by use ceramsite in comparison to other filter materials? 10%?, 20?..please provide a value.
Line 47: “we present..”, please don’t use the first person in the entire paper.
Line 47-56: Are you sure about this paragraph?....in my opinion, this paragraph is wrong. Please, correct.
Methods: Please provide the strategy for operation and monitoring. Add, Hydraulic retention time, are you working with real sewage or laboratory sewage?, What is the hydraulic loading rate?. Are you taking samples each week, every two weeks?. What are your analytical methods for doing COD, ammonia, and phosphorus?
Methods: You told me in abstract, line 17 “….different start-up…”, but in methods you didn´t tell anything about the strategies employed in your work
Line 112-118 and Figure 2: please put the text with the description of the figure prior to show the figure. Same commentary for the rest of figures and tables.
Line 120-126: Why do you use double space?
Line 121: The increasing in the properties described by you is a consequence of...? Your sentence is unclear.
Line 155-156: Reading this sentence is unclear the different stages that you are trying to describe. Please explain in a better way.
Line 172: Please add a citation for supporting your sentence.
Line 177: What is the first stage?...
Line 181: References for this standard.
Line 199: What studies are you talking about?. Remember, this a scientific paper, so, you have to put the references of the different studies.
Line 208-209: The text “Error!, Reference source not found”. Please edit in a good way your paper before sending.
Line 209: How much is an excesive HRT for a constructed wetland?, 5 d, 10 d, 100 d?
Line 215-216. Previously, you did not tell me anything about different HRT. Now you are talking about that. You must mention this situation previously.
Line 261: Similar to the previous comment. You did not tell me anything about variations in height.
Please, for the next time, check in a carefully way your text prior to sending.
Author Response
We are grateful for having the chance to improve our manuscript entitled “Preparation of Ceramsite Based on Waterworks Sludge and Its Application in Constructed Wetlands” (Manuscript ID: ijerph-520349). We would also like to express our great appreciation and sincere thanks to the reviewers for thoroughly reviewing our manuscript and making many constructive and positive comments. We have addressed all the comments and revised the manuscript accordingly. With this letter, we attach the detailed responses to all comments

Reviewer 3 Report
It is an interesting paper, but it has some major flaws. Materials and methods section is not complete, and many things are missing. In the Results I found some trials that weren't presented in M&Ms. Also, some results need to be better discussed, since some explanations given are not suitable.
Line 27-28 Where did it reach 6.25×106 tons? In China, Asia, world?
What were objectives of your work? What did you want to study?
Lines 47-56 This part does not belong to the Introduction. Part of it Materials and methods and another part results.
Line 63 How did you enable upward flow? Why did you use upflow and not downflow?
Line 64 The total filling height was 750 mm, out of which 600 were ceramsite and 100 cobbles. Where is the rest of 50 mm?
Figure 1 I would suggest adding the heights to the figure, it would be more clear.
Line 78 How did you collect the substrate?
How long was your experimental period? What kind of wastewater did you use? How did you analyse samples?
Line 87 This is not how the formulas should be written.
You should always introduce the figure in the text before you actually put the figure. Please do that for all the figures.
Line 130 How did you get nitrogen adsorption/desorption isotherms? Through an experimental activity? If so, you should add that part to the Materials part.
Figure 4 Why such a big difference in the influent strength in the first 30 days and after? You didn't mention that in M&Ms.
Line 182 Reference?
Line 199 What studies?
Line 229 Full play?
Line 234-236 Any reference for this statement?
Line 257 Did you mention this trial in M&Ms? Or maybe did you use outlets in different depths of the column to test the filling height influence? I guess not because here you say you tested also 15 cm depth and the lowest outlet was 30 cm. You cannot just say that you tested something and the result was x. All the experiments you conducted HAVE to be explained in the Materials and Methods. Also, did you test those heights only once or more times. If it was only once, maybe it is not the real results because of some condition that happened only that time. If you tested several effluents, give average value with standard deviation and number of samples.
Line 258 Not clear at all. What aforementioned wastewater?
Figure 6 Here you have depths 10, 20, 30 and so on, while in the text you wrote 15, 30, 45. Which one is correct?
Line 262 I don’t see from the figure that removal rate of COD rose continuously from 0 to 30 cm. First of all, there is no value for 0 cm. Second of all, it rose also beyond 30 cm. The explanation you gave is not reflecting the results. If the COD removal concentrated only in the bottom half, then there wouldn’t be difference between higher depths. Did you conduct statistical tests? Were these differences significant or not?
Line 282-283 Can you support this statement with a reference? TP is usually removed by physical processes (eg. adsorption), while removal through biomass uptake is minor. Here, if I understood well, you suggest that TP was removed mainly though biomass?
Line 284 CWs are an advanced wastewater treatment process? What do you mean by advanced?
Line 291-292 Reference?
Line 297 I wouldn’t call this sudden changes or impact of increasing NH3-N concentration. The concentrations you gave in Figure 7 (both influent and effluent) are practically the same.
Line 305 What is Alpha diversity?
Line 309 What are all these parameters? I have never heard of them. You should give a theoretical explanation before using them.
Line 332 How do you see that the ceramsite-constructed wetland with a water supply sludge had certain advantages in the total amount of denitrification microorganisms?
In Line 47 you say that you used ‘’ceramsite made from waterworks sludge’’, while in Line 332 you say ‘’ceramsite-constructed wetland with a water supply sludge’’. Were they two different substrates or one substrate? Please make this clear from the beginning, there were also other places in the paper where I had the same doubt.
Conclusions need to be rewritten. It is an abstract, and not the conclusion. In Conclusions part you should give an overall conclusions made from the results you obtained. Can that specific design be used for efficient wastewater treatment? What are the advantages and what are disadvantages? How can the design be changed in order to be more efficient?
Author Response
We are grateful for having the chance to improve our manuscript entitled “Preparation of Ceramsite Based on Waterworks Sludge and Its Application in Constructed Wetlands” (Manuscript ID: ijerph-520349). We would also like to express our great appreciation and sincere thanks to the reviewers for thoroughly reviewing our manuscript and making many constructive and positive comments. We have addressed all the comments and revised the manuscript accordingly. With this letter, we attach the detailed responses to all comments.

Round 2
Reviewer 1 Report
The authors responded satisfactorily to the comments. It is required to verify the English text of the added sections.

Author Response
Response to Reviewer 1 Comments
Dear reviewer,
We are grateful for having the chance to improve our manuscript entitled “Preparation of Ceramsite Based on Waterworks Sludge and Its Application as Matrix in Constructed Wetlands” (Manuscript ID: ijerph-520349). We would also like to express our great appreciation and sincere thanks to you for thoroughly reviewing our manuscript and making many constructive and positive comments. We have addressed all the comments and revised the manuscript accordingly. With this letter, we attach the detailed responses to all comments raised. We also provide the revised manuscript in both marked and clean version.
Thank you for your kind attention on our manuscript! My co-authors and I look forward to your decision.
Yours sincerely,
Yaning Wang
Point: The authors responded satisfactorily to the comments. It is required to verify the English text of the added sections.
We appreciated you for the positive evaluation and we addressed all the comments in the following.
Response 1:
“wetlandas” has been changed to “wetlands” in line 50.
Response 2:
“After the construction of the constructed wetland, it began to run.” has been deleted in line 71.
Response 3:
“With the operation of wetland system becoming more and more stable, the quality of effluent water became more and more stable.” has been changed to “After the start-up and adaptation period, the effluent quality was stable so that sampling and determination were adjusted to twice a week.” in line 73-74.
Response 4:
“When it comes to the experiment about different HRTs,” has been changed to “Different HRTs of sewage in the constructed wetland was controlled by adjusting the inflow rate” in line 84.
Reviewer 2 Report
Dear authors. I feel that your paper has improved. However, in my opinion, the short time in your response show me that you didn't take enough time for editing the final version (as it was previously commented). For example,
What is the "n" for Table 1?.
Response to point 5 is wrong. You did not answer me that I asked
I feel that your experimental design is very confusing and you employed many variables in a short time for your experiment. Now, that I can see your materials and methods in a better way, I am wondering, Why the authors did not use any statistical analysis?
Many of your answers were referring to the answer to point 6. Respectfully, I think you have to work a little more in a detalied way your answers and to give more explanations and improve your results.
Figure 6, You did not have variations?...you only show bar chart but not errors. This bar chart is only for one sample?
In addition, in my previous revision, I told you that your experiment is not a constructed wetland. The reason, You did not use plants. Remember, the constructed wetland concept involves plants. You are working on a proposal medium for a constructed wetland. That's it. Please reconsider this recommendation on your title.
Author Response
Response to Reviewer 2 Comments
Dear reviewer,
We are grateful for having the chance to improve our manuscript entitled “Preparation of Ceramsite Based on Waterworks Sludge and Its Application as Matrix in Constructed Wetlands” (Manuscript ID: ijerph-520349). We would also like to express our great appreciation and sincere thanks to you for thoroughly reviewing our manuscript and making many constructive and positive comments. We have addressed all the comments and revised the manuscript accordingly. With this letter, we attach the detailed responses to all comments raised. We also provide the revised manuscript in both marked and clean version.
Thank you for your kind attention on our manuscript! My co-authors and I look forward to your decision.
Yours sincerely,
Yaning Wang
Point 1: Dear authors. I feel that your paper has improved. However, in my opinion, the short time in your response show me that you didn't take enough time for editing the final version (as it was previously commented). For example, What is the "n" for Table 1?
Response 1:We appreciated you for the positive evaluation and this time we has used more time for editing. Table 1 was shown as follows and we suppose there was no “n”?
Table 1. Water quality index of tail water from Nanjing JX Sewage Treatment Plant.
pH | COD (mg/L) | NH3-N (mg/L) | NO3-N (mg/L) | NO2-N (mg/L) | TN (mg/L) | TP (mg/L) |
7.4 | 38.24 | 2.19 | 9.35 | 0.67 | 13.06 | 0.35 |
Point 2:Response to point 5 is wrong. You did not answer me that I asked
Response 2: We felt very sorry for not answering your question. But in our opinion we suppose the figure had no problem. We actually followed the figure and finished the experiment.
This was the real picture. Please see the figure in the attachment .
Point 3:I feel that your experimental design is very confusing and you employed many variables in a short time for your experiment. Now, that I can see your materials and methods in a better way, I am wondering, Why the authors did not use any statistical analysis?
Response 3: We felt very sorry for not use any statistical analysis and thank you very much for your suggestion. The statistical analysis has been added in Figure 6.
Please see the figure in the attachment .
We have added the variation and the error bar in this figure.
Point 4:Many of your answers were referring to the answer to point 6. Respectfully, I think you have to work a little more in a detalied way your answers and to give more explanations and improve your results. Figure 6, You did not have variations?...you only show bar chart but not errors. This bar chart is only for one sample?
Response 4: Sorry for not showing variations and these have been added. The bar chart was the mean value for a set of numbers and the error bar shown the standard deviation of these numbers.
Point 5:In addition, in my previous revision, I told you that your experiment is not a constructed wetland. The reason, You did not use plants. Remember, the constructed wetland concept involves plants. You are working on a proposal medium for a constructed wetland. That's it. Please reconsider this recommendation on your title.
Response 5:
Thank you very much for your suggestion and it is a very important point.
The title has been changed to “Preparation of Ceramsite Based on Waterworks Sludge and Its Application as Matrix in Constructed Wetlands”.

Reviewer 3 Report
I think that after the authors have applied my comments from below, the paper will be scientifically sound for publishing. However, English needs quite some polishing and I would recommend to the authors to take it seriously since some parts of the paper are really difficult to understand.
Response 4: I highly doubt that most of the real constructed wetlands use bottom water distribution and upward flow.
Response 7: You never mentioned in the paper that you had a few systems that were working in parallel. That is an important detail, and cannot be omitted!
Response 8: It is still not clear how long was your experimental period. You explained the duration of the start-up period, but not that one of the experimental period.
Response 17: What do you mean by choosing a group of best numbers? You have to report all the numbers, you cannot just choose the ones you like and discard those you don’t like.
Response 18: I still don’t understand what wastewater are you talking about.
Response 19: I don’t see any change in Figure 6. The way it is now it seems that the removal rate of TP at 50 cm height was 75%. Please change the figure so that the real results are given in a clear way. There is no need for both these figures, they give the same results.
Response 22: Advanced does not mean not traditional. Please correct.
Response 24: What are discharge standard values? You should give them in the paper. Why did you decide to treat that strength of wastewater? It is already quite low, why does it need further treatment?
Response 25: Give that explanation in the paper and support it with a reference, you cannot assume that everybody know about that parameter.
Author Response
Response to Reviewer 3 Comments
Dear reviewer,
We are grateful for having the chance to improve our manuscript entitled “Preparation of Ceramsite Based on Waterworks Sludge and Its Application as Matrix in Constructed Wetlands” (Manuscript ID: ijerph-520349). We would also like to express our great appreciation and sincere thanks to you for thoroughly reviewing our manuscript and making many constructive and positive comments. We have addressed all the comments and revised the manuscript accordingly. With this letter, we attach the detailed responses to all comments raised. We also provide the revised manuscript in both marked and clean version.
Thank you for your kind attention on our manuscript! My co-authors and I look forward to your decision.
Yours sincerely,
Yaning Wang
Point 1: I think that after the authors have applied my comments from below, the paper will be scientifically sound for publishing. However, English needs quite some polishing and I would recommend to the authors to take it seriously since some parts of the paper are really difficult to understand.
Response 4: I highly doubt that most of the real constructed wetlands use bottom
Response 1: We appreciated you for the positive evaluation. The aim of the upward flow direction was to improve the internal environment of matrix, make oxygen more sufficient to promote the proliferation of aerobic microorganisms, facilitate better use of wetland matrix to remove pollutants, and slow down matrix blockage [13].
This has been added in line 66-69.
And there is the reference:
[13] Zhou, L.C. Study on the development and assessment of a novel vertical flow constructed wetland system using ceramsiteas substrate. Xi’an University of Architecture and Technology 2013.
Point 2:Response 7: You never mentioned in the paper that you had a few systems that were working in parallel. That is an important detail, and cannot be omitted!
Please see the figure in the attachment.
Response 2: We felt very sorry for omitting this and thank you very much for your suggestion.
There were actually two systems working in parallel and this was the real picture.
The description has been added in the line 71.
Point 3:Response 8: It is still not clear how long was your experimental period. You explained the duration of the start-up period, but not that one of the experimental period.
Response 3: Thank you very much for your suggestion. Experimental period was actually running time of the constructed wetland in Figure 4,5,7.
“When the experimental period was from 0-50 d, which was the start-up period, removal rate of COD, NH3-N and TP were tested. Then from 51-110 d, different HRTs were selected and the removal rate of COD, NH3-N and TP were again tested during this period. Finally from 111-130 d, removal rate of NH3-N was tested when the influent NH3-N exceeded the standard.”
This has been added from line 87-line 90.
Point 4:Response 17: What do you mean by choosing a group of best numbers? You have to report all the numbers, you cannot just choose the ones you like and discard those you don’t like.
Please see the figure in the attachment.
Response 4: We felt very sorry for not making this clear and thank you very much for your suggestion. The bar chart was the mean value for a set of numbers and the error bar shown the standard deviation of these numbers.
Point 5:Response 18: I still don’t understand what wastewater are you talking about.
Response 5: We felt very sorry for not clearly showing the wastewater and thank you very much for your suggestion. The wastewater was the water shown in table 1.
Table 1. Water quality index of tail water from Nanjing JX Sewage Treatment Plant.
pH | COD (mg/L) | NH3-N (mg/L) | NO3-N (mg/L) | NO2-N (mg/L) | TN (mg/L) | TP (mg/L) |
7.4 | 38.24 | 2.19 | 9.35 | 0.67 | 13.06 | 0.35 |
Point 6: Response 19: I don’t see any change in Figure 6. The way it is now it seems that the removal rate of TP at 50 cm height was 75%. Please change the figure so that the real results are given in a clear way. There is no need for both these figures, they give the same results.
Response 6: Thank you very much for your suggestions and the figure has been changed to that shown in response 4. The bar chart was the mean value and the error bar shown the standard deviation. We hope this time the figure can show the numbers in a clear way.
Point 7:Response 22: Advanced does not mean not traditional. Please correct.
Response 7: Sorry for our misunderstanding, “As an advanced treatment process of wastewater, constructed wetlands play an important role in ensuring the discharged wastewater meets the standards and maintains its water quality.” has been deleted.
Point 8:Response 24: What are discharge standard values? You should give them in the paper. Why did you decide to treat that strength of wastewater? It is already quite low, why does it need further treatment?
Response 8: We wanted to prove that the constructed wetland can treat tailwater from wastewater treatment plant to meet Class IV Water Standard of Surface Water Environmental Quality Standard (GB3838-2002).
The reference has been added “[19] The state environmental protection sdministration and the state administration of quality supervision, inspection and quarantine promulgated surface water environmental quality standards. China's Environmental Protection Industry 2002, 6, 5-6.” which showed the specific values of Class IV Water Standard of Surface Water Environmental Quality Standard (GB3838-2002).
Point 9:Response 25: Give that explanation in the paper and support it with a reference, you cannot assume that everybody know about that parameter.
The authors responded satisfactorily to the comments. It is required to verify the English text of the added sections.
Response 9:
We felt very sorry for not explaining this well and thank you very much for your suggestion.
OTU means operational taxonomic units and it is used to observe diversity richness. ACE and Chao1 are richness estimators which are used to estimate OTU richness. Shannon, Simpson and the Overage are diversity indices which are used to estimated sample coverage.
The reference “[30] Li, T.; Long, M.; Gatesoupe, F.J.; Zhang, Q.; Li, A.; Gong, X. Comparative Analysis of the Intestinal Bacterial Communities in Different Species of Carp by Pyrosequencing. Microbial Ecology 2014, 69(1), 25–36. ” has been added after the explanation “Chao1 and the ACE analysis index can be used to calculate and analyse the distribution abundance of the microbial communities. Shannon, Simpson and the Overage index can analyse the diversity of the microbial community distribution [30].”
Round 3
Reviewer 2 Report
My only comment, I miss a deeper statistical analysis because you are working with different variables
Author Response
Point 1: My only comment, I miss a deeper statistical analysis because you are working with different variables
Response 1: Thank you very much for your suggestion,
We have added “Then, with an HRT of 2 d, sampling and determination were adjusted to be taken every two days and the removal rates of COD, NH3-N and TP were calculated at 15 cm, 30 cm, 45 cm and 60 cm, respectively. At each filling height, 10 samples were taken totally during 20 d and the mean value was calculated.” in the methods part in line 99-101 to make the experimental design more clear.
“Although the data fluctuated above and below the mean value, the deviation was small and the overall trend can still show the difference between COD, NH3-N and TP clearly.” has been added in line 280-281.
Reviewer 3 Report
The response of the authors to my last round of comments show that they did not consider and report many important things, and therefore it makes me question how many other things that I could not see were not reported. Experimental design and presentation of results is still quite confusing, and English still needs to be improved. Although the article has some potential, I do not feel comfortable accepting it in this form. I would suggest to the authors to take some time and reorganise and rewrite the paper in order for it to be clear for the future reviewers and readers.
Author Response
Response 1: We felt very sorry for not making this clear and thank you very much for your suggestion. We suppose we have really reported everything now.
We have added “Then, with an HRT of 2 d, sampling and determination were adjusted to be taken every two days and the removal rates of COD, NH3-N and TP were calculated at 15 cm, 30 cm, 45 cm and 60 cm, respectively. At each filling height, 10 samples were taken totally during 20 d and the mean value was calculated.” in the methods part in line 99-101 to make the experimental design more clear.
The bar chart was the mean value and the error bar shown the standard deviation of these numbers. The error bar were shown in the figure to make the presentation of results more clear. “Although the data fluctuated above and below the mean value, the deviation was small and the overall trend can still show the difference between COD, NH3-N and TP clearly.” has been added in line 280-281.
